



# ISWFM-NSCS v2.0: advancing the internal solitary wave forecasting model with background currents and horizontally inhomogeneous stratifications

Yankun Gong[1], Xueen Chen[2], Jiexin Xu[1], Zhiwu Chen[1], Qingyou He[1,3], Ruixiang Zhao[4], Xiao-Hua Zhu[4], Shuqun Cai[1,5]

[1]State Key Laboratory of Tropical Oceanography, South China Sea Institute of Oceanology, Chinese Academy of Sciences, Guangzhou 510301, China
[2]College of Oceanic and Atmospheric Sciences, Ocean University of China, Qingdao, 266100, China
[3]Guangdong Key Lab of Ocean Remote Sensing, South China Sea Institute of Oceanology, Chinese Academy of Sciences, Guangzhou 510301, China
[4]State Key Laboratory of Satellite Ocean Environment Dynamics, Second Institute of Oceanography, Ministry of Natural Resources, Hangzhou, 310012, China
[5]University of Chinese Academy of Sciences, Beijing, 100049, China

*Correspondence to*: Shuqun Cai (caisq@scsio.ac.cn)

**Abstract.** A new version of an internal solitary wave (ISW) model, the Internal Solitary Wave Forecasting Model-Northern South China Sea version 2.0 (ISWFM-NSCS v2.0), is presented. The background currents and horizontally inhomogeneous stratifications are implemented in ISWFM-NSCS v2.0 to better reproduce ISW properties, including arrival time, mode-1 wave amplitude, wave-induced velocity, characteristic half-width and propagation direction. Optimized viscosity and diffusivity coefficients (i.e., $1.0 \times 10^{-2}$ $m^2$ $s^{-1}$ in horizontal and $1.0 \times 10^{-5}$ $m^2$ $s^{-1}$ in vertical) are also introduced in ISWFM-NSCS v2.0 to maintain stable stratifications within the model domain, thereby prolonging the model's valid forecasting period. A mooring station around the Dongsha Atoll is presented for model evaluation and numbers of sensitivity experiments are implemented to illustrate the individual effect of the major updates. In comparison with ISWFM-NSCS v1.0, ISWFM-NSCS v2.0 significantly enhances model accuracy in forecasting ISW characteristics, with a 37% improvement in arrival time, a 34% improvement in mode-1 wave amplitude, a 25% improvement in wave-induced velocity, and an 85% improvement in half-width.

## 1 Introduction

For several decades, numerical modelling has been regarded as a crucial methodology for investigating internal solitary waves (ISWs) in the global ocean. To our knowledge, one of the earliest ISW model, Korteweg-de Vries equation, dates back to last century (Benny, 1966; Gear and Grimshaw, 1983; Holloway, 1987), utilizing a two-layer approximation. Since the northern South China Sea (NSCS) has been known as one of the most active regions for ISWs, a variety of two-dimensional (2D) models have gradually emerged in the 21st century (Du et al., 2008), driven by advancements in high-





performance computing. As reviewed by Simmons et al. (2011), these models have continuously improved in capability, evolving from 2D to 3D and from hydrostatic to non-hydrostatic formulations. Subsequently, a substantial number of 3D realistic ISW models have been developed in non-hydrostatic mode, including SUNTANS (Zhang et al., 2011), MITgcm
(Alford et al., 2015) and FVCOM (Lai et al., 2019).

An ISW forecasting model in the NSCS (Gong et al., 2023), called ISWFM-NSCS v1.0 hereafter, is one of the 3D realistic ISW models developed by a primitive equation ocean solver (MITgcm, Marshall et al., 1997). ISWFM-NSCS v1.0 has demonstrated robust performance in characterizing various wave properties within 15 modal days, as evidenced by comparisons to field observations and satellite images. However, as discussed in Gong et al. (2023), there remains potential
for enhancement in the coefficient configurations, initial conditions and boundary conditions.

Firstly, the valid forecasting period in ISWFM-NSCS v1.0 is limited to 15 days, as the stratifications significantly weaken beyond this time frame. Viscosity and diffusivity parameters determine the extent of mixing and dissipation of energy in the model. Higher coefficients increase the damping of internal waves, leading to smoother wave fields and reduced wave amplitudes. This can help in preventing numerical instabilities but might underestimate the wave energy and dynamics (Legg
and Huijts, 2006). In contrast, low-valued coefficients might lead to numerical noise and spurious oscillations (Álvarez et al., 2019). Therefore, optimizing the settings for viscosity and diffusivity coefficients is essential for maintaining numerical stability in high-resolution models (Nagai and Hibiya, 2015; Stewart et al., 2017) and extending the valid forecasting period (e.g., 30 days or more).

Secondly, initial stratification profiles (temperature and salinity) in ISWFM-NSCS v1.0 are horizontally homogeneous.
However, previous studies (e.g., Centurioni et al., 2004; Chao et al., 2007) have observed that the thermocline in the Luzon Strait rises in the west due to the influence of the northward-flowing Kuroshio. Zheng et al. (2007) proposed that the deepening of the thermocline towards the east could potentially hinder the development of eastward-moving solitons in the Pacific.. Additionally, 2D idealized model simulations by Shaw et al. (2009) and Buijsman et al. (2010) demonstrated that a sharper and shallower thermocline on the western side of the Luzon Strait, compared to the eastern side, leads to the
generation of larger westward-propagating solitons. While these studies shed light on how horizontally inhomogeneous stratification affects the east-west asymmetry of internal solitary waves, their findings have yet to be verified using a 3D realistic model.

Thirdly, ISWFM-NSCS v1.0 does not account for background currents, which are a crucial factor in ISW dynamics. While it is well-established that background stratification and currents significantly affect the characteristics and behaviour
of ISWs (DeCarlo et al., 2015; Li et al., 2016), the effects of more complex dynamical motions, such as oceanic currents and mesoscale eddies, remain inadequately explored. Specifically, the effects of the Kuroshio (Caruso et al., 2006) and mesoscale eddies (Xie et al., 2015) on ISWs have not yet been examined using a 3D realistic model. The absence of such considerations in ISWFM-NSCS v1.0 suggests a gap in our understanding, as incorporating these dynamic elements could provide deeper insights into how ISWs interact with background currents.



In this work, we present an updated version of a high-performance model for predicting ISWs in the NSCS (called ISWFM-NSCS v2.0 hereafter) and evaluate the roles of optimized turbulence configurations, horizontally inhomogeneous stratifications, and background currents in precisely forecasting ISWs by comparing with numbers of sensitivity numerical experiments. The structure of the manuscript is outlined as follows. A description of the model is given and the major updates made from ISWFM-NSCS v1.0 to ISWFM-NSCS v2.0 are summarized in Sect. 2. The updated model results and
corresponding calibrations are detailed in Sect. 3. Moreover, we provide a quantitative analysis to demonstrate the roles of optimized viscosity and diffusivity, horizontally inhomogeneous stratifications, and background currents in the ISWFM-NSCS v2.0 in Sect. 4. Conclusions follow in Sect. 5.

## 2 Description of the model and major updates

As ISWFM-NSCS v1.0 has been described in Gong et al. (2023), only the main concepts are reviewed here. In ISWFM-
NSCS, a 3D realistic non-hydrostatic oceanic solver MITgcm (Marshall et al., 1997) is employed to reproduce generation, propagation and dissipation processes of ISWs in the NSCS. As discussed in ISWFM-NSCS v1.0, the horizontal cell size is set to 500 m, with 90 vertical layers (i.e., from 5 m at the sea surface to 120 m at the bottom). A time step of 10 sec ensures compliance with the Courant-Friedrichs-Lewy (CFL) conditions. Horizontally homogeneous temperature and salinity profiles are initialized by a climatology WOA 2018 dataset (https://www.ncei.noaa.gov/access/world-ocean-atlas-2018/, last
access: 10 July 2024) and the model bathymetry is obtained by interpolating the GEBCO dataset (https://www.gebco.net/data_and_products/gridded_bathymetry_data, last access: 10 July 2024). Eight primary barotropic tidal constituents, extracted from TPXO8-atlas dataset (Egbert and Erofeeva, 2002), are applied at each lateral boundary, with a 25 km wide sponge layer absorbing internal wave energy (Zhang et al., 2011). However, background currents and eddies have not been taken into account in ISWFM-NSCS v1.0. To mitigate grid-scale instability, constant turbulent
parameters (including viscosity and diffusivity) are applied as follows: $A_h = 0.5$ m$^2$ s$^{-1}$; $A_v = 5.0 \times 10^{-3}$ m$^2$ s$^{-1}$; $K_h = 0.5$ m$^2$ s$^{-1}$; $K_v = 5.0 \times 10^{-3}$ m$^2$ s$^{-1}$.

In comparison to field observations and satellite imagery, a test case applying the ISWFM-NSCS v1.0 shows well performance in reproducing ISW properties within first 10 model days. Specifically, the root mean square deviations (RMSD) of arrival time, maximum vertical wave amplitude and baroclinic velocity are 0.71 h, 37.27 m and, 0.41 m s$^{-1}$, respectively.
However, the stratification profiles gradually weaken with descending thermocline depth after the 10[th] model days, which limit the effective duration of the forecasting model. As discussed in ISWFM-NSCS v1.0, the prediction accuracy may be improved to take into account background currents (Xie et al., 2015) and horizontally inhomogeneous stratifications (Buijsman et al., 2010). Then, this section describes the major updates in comparison with ISWFM-NSCS v1.0 (marked in red in Fig. 1).



## 2.1 Optimizations of viscosity and diffusivity

In ISWFM-NSCS v1.0, high values of eddy viscosity and diffusivity coefficients (i.e., 0.5 $m^2\,s^{-1}$ in horizontal and $5.0\times10^{-3}$ $m^2\,s^{-1}$ in vertical) are empirically selected to be sufficient to eliminate grid-scale noise in the velocity and mixing fields (Legg and Huijts, 2006). However, these values generally weaken the background stratifications within the entire model domain, in particular after two weeks, potentially dampening ISW amplitudes and underestimating the wave nonlinearity. Hence, the valid forecasting duration of ISWFM-NSCS v1.0 is less than 15 days, due to the weakening stratification.

Intensified diffusivity was observed exceeding the order of $10^{-3}$ $m^2\,s^{-1}$ in magnitude in the NSCS (Sun et al., 2016), which were caused by strong internal tides and winds in winter. Nonetheless, more microstructure measurements and finescale parameterization assessed the diapycnal (vertical) diffusivity as the order of $10^{-5}$ $m^2\,s^{-1}$ in summer (i.e., Shang et al., 2017). To be consistent with in-situ observations, vertical diffusivity coefficients is also set as $1.0\times10^{-5}$ $m^2\,s^{-1}$ (see Fig. 1) in ISWFM-NSCS v2.0, following Vlasenko et al. (2010). In addition, horizontal diffusivity coefficient is set on the order of $1.0\times10^{-2}$ $m^2\,s^{-1}$, a value that has been validated as appropriate by Gent (2011).

Horizontal turbulence in the ocean is driven by large-scale processes like ocean currents and mesoscale eddies (Large et al., 1994). To represent the effective mixing, the horizontal eddy viscosity is set as $1.0\times10^{-2}$ $m^2\,s^{-1}$, as suggested by Gent and McWilliams (1986). In comparison with horizontal processes, vertical turbulence is mainly influenced by smaller-scale processes, such as internal waves, shear instabilities and convection. As a result, the vertical eddy viscosity is set much lower in ISWFM-NSCS v2.0, namely $1.0\times10^{-5}$ $m^2\,s^{-1}$, following Mellor and Yamada (1982).

## 2.2 Horizontally inhomogeneous stratifications

Numerous satellite images concluded the east-west asymmetric characteristics of ISWs in the Luzon Strait in the NSCS (e.g., Jackson and Apel, 2004; Alford et al., 2015). In previous literatures, the asymmetry of ISWs is a multifaceted phenomenon influenced by asymmetric barotropic tides, water depth differences between the NSCS and Pacific Ocean, westward thermocline shoaling related to the Kuroshio current, and internal tide resonance in a double ridge configuration. In ISWFM-NSCS v1.0, three factors have been considered except for the east–west gradients in the thermocline.

The westward shoaling of the thermocline is associated with the northward-flowing Kuroshio current, which is centered between the west and east ridges, as evidenced by drifter observations (Centurioni et al., 2004) and model results (Chao et al., 2007). Based on a nonhydrostatic ROMS model, Buijsman et al. (2010) demonstrated that westward-propagating solitons are 28% larger than eastward-propagating solitons, due to the inhomogeneous thermocline, which is secondary to the effect of a deeper Pacific Ocean. In addition, Zheng et al. (2007) argued that the eastward deepening of the thermocline might inhibit the formation of eastward-propagating solitons in the Pacific.

Aforementioned evidences highlight the significant roles of thermocline structure in shaping the east-west asymmetric characteristics of ISWs in the Luzon Strait. Therefore, horizontally inhomogeneous stratifications are conducted as the initial





conditions (Figs. 2a and 2b) in ISWFM-NSCS v2.0. Temperature and salinity profiles are both extracted from the global HYCOM re-analysis dataset (https://www.hycom.org/, last access: 12 July 2024).

## 2.3 Background currents and eddies

In the NSCS, background circulations such as the Kuroshio Current play a crucial role in influencing behaviour and characteristics of ISWs (DeCarlo et al., 2015; Li et al., 2016). The Kuroshio, a major western boundary current, brings warm and salty waters into the NSCS, thereby significantly affecting the local hydrography and stratification (Hu et al., 2000). When ISWs encounter the Kuroshio, the interaction can alter the propagation speed and direction of ISWs (Alford et al., 2010). Specifically, a strong north-westward flow of the Kuroshio (looping or leaking type) can accelerate ISWs traveling in the same direction, enhancing their amplitude and nonlinearity. Conversely, ISWs moving against the Kuroshio current can experience deceleration and reduced amplitude, affecting their arrival time and energy. This dynamic interaction affects temporal and spatial characteristics of ISWs, impacting the mixing processes and energy distribution in the NSCS (Xie et al., 2021).

Mesoscale eddies are another ubiquitous phenomenon (Chelton et al., 2011), which are dynamically important in modulating currents and temperature in the NSCS. Their interaction with ISWs can be expected to happen frequently in the deep basin (Huang et al., 2017). Numerical simulations by Xie et al. (2015) demonstrated that mesoscale eddies can redistribute the energy of ISWs along their wave fronts. In regions where energy is focused, the amplitudes of ISWs tend to increase, while in spreading regions, they decrease. Previous observational studies (Park and Farmer, 2013; Li et al., 2016) also showed that mesoscale structures can substantially distort the propagation paths of ISWs in the NSCS, leading to dramatic changes in wave amplitude at fixed locations.

Given the importance of background circulations and mesoscale eddies to ISW properties, background currents are not only added as the initial condition (Fig. 2c), but also continuously imposed at the four lateral boundaries (Fig. 3) over time in ISWFM-NSCS v2.0. Zonal and meridional velocity data are also extracted from the global HYCOM re-analysis dataset (https://www.hycom.org/, last access: 12 July 2024).

## 3 Model results and calibrations

Following the aforementioned updates, a reference test case (i.e., control run, EXP. 1) is launched since 05 August 2014 and run for 30 days, including two spring-neap cycles. The model runs with a sampling rate of 1 h in the whole domain, and also a sampling rate of 1 min at a targeted location for comparing to field observations.

Then, the model performance is evaluated in three stages: first, by comparing the background current field with the HYCOM reanalysis dataset; second, by comparing the spatial characteristics of ISWs with satellite imagery; and third, by comparing 28 ISW properties (including arrival time, maximum vertical amplitudes, baroclinic velocities, and characteristic half-widths) with field observational data from the targeted mooring DS.



## 3.1 Comparison with HYCOM reanalysis dataset

To evaluate the model accuracy in reproducing the correct background current field, we run an extra 3D model (EXP. 0) with the same configurations as EXP. 1, but exclude the surface tide forcing at four lateral boundaries (see details in Table 1). Furthermore, both the horizontal resolution (reduced from 500 m to 1/12°) and the vertical resolution (reduced from 90 layers to 40 layers) are adjusted downward in EXP. 0 to maintain consistency with the HYCOM reanalysis dataset.

As shown in Fig. 4, we depict four snapshots with a 10-day interval from 05 August to 04 September for the HYCOM reanalysis dataset and the EXP. 0 model results, respectively. Note that main flow patterns are marked with red arrows. It is clear that in both HYCOM reanalysis dataset and EXP. 0 results, the Kuroshio flows northward from the east side of Philippine to the east side of Taiwan Island (Figs. 4a and 4b) with a leaking-pattern intrusion after 20 days (Figs. 4c and 4g). In addition, an anticyclonic eddy is also reproduced in the model at the east side of Luzon Strait after 10 days (Figs. 4b and 4f). However, the model (EXP. 0) might omit a small eddy at the west side of Taiwan Island near the north boundary at the end (Fig. 4h), which is likely to generate remotely and propagate into the model domain. But overall, the model can correctly reproduce background current field in the NSCS, including the Kuroshio and mesoscale eddies.

## 3.2 Comparison with satellite images

In addition to validating the model's ability in background current regime, we subsequently examine the control run (EXP. 1) in ISW field via a comparison between the model with MODIS imagery (available in the NASA Worldview website, https://worldview.earthdata.nasa.gov, last access: 19 July 2024). Given the model's one-hour sampling rate, we choice four closest snapshots of sea surface height gradients for comparison with MODIS imagery (Fig. 5). Note that detailed approaches to compute sea surface height gradients can be found in Gong et al. (2023).

Figs. 5a (05:00 UTC, 14 August) and 5b (05:15 UTC, 14 August) both illustrate two consecutive ISWs (labelled as IW1 and IW2) separated by approximately 120 km. The predicted curvatures, lengths, and positions of IW1 and IW2 exhibit a high degree of consistency with the corresponding features observed in the satellite imagery. Nevertheless, the numerical simulations also reveal two additional ISWs on the continental slope (Fig. 5a), which are obscured in the MODIS-Aqua image from 14 August due to cloud cover (Fig. 5b). On 15 August, the cloud cover cleared, allowing the MODIS-Terra image to capture a clearer depiction of the ISWs (Fig. 5d). The three ISWs shown in Figs. 5c and 5d are located in close proximity and exhibit similar wave crestline lengths extending from the Luzon Strait to the continental slope. Additionally, in shallower waters, the simulated IW1 shows an ISW packet with secondary waves, a feature also observed in the satellite imagery.

Throughout the extended 15-day forecast period in ISWFM-NSCS v2.0, EXP. 1 continues to demonstrate strong performance in depicting spatial distributions of ISWs (Figs. 5e – 5h). Specifically, satellite-observed shallowing and diffracting processes around the Dongsha Atoll at 05:25 UTC on 28 August (Fig. 5f) are clearly captured by the model at 05:00 UTC on 28 August (Fig. 5e). After ISWs impact the Dongsha Atoll (IW4), their wave crests are divided into two



branches (IW2 and IW3). The lengths of the wave crests shorten as they bypass the atoll and continue to propagate westward,
eventually reconverging behind the island (IW1). It is worth mentioning that the model and the satellite observations remain
consistent even on the 25th day of the forecast period (31 August, Figs. 5g and 5h).

Given that the control run does not account for wind effects above the sea surface, some subtle differences in wave
characteristics remain. Nevertheless, the model effectively illustrates the spatial features of ISWs in the NSCS, as evidenced
by comparisons with the MODIS imagery.

## 3.3 Comparison with field observations

To conduct a more detailed evaluation of the model's accuracy in predicting ISWs, we incorporate field observations from
the Dongsha station (see details in Gong et al., 2023, hereafter DS station). We examine the vertical structure and arrival
time of ISWs after their passage through the deep basin by plotting temperature and baroclinic velocities (or wave-induced
velocities) for the periods of 08 to 14 August and 25 to 31 August, respectively. For clarity, Fig. 6 only displays the
comparison for the upper 900 m, encompassing the primary wave-induced temperature fluctuations.

Throughout the initial 15-day period, both the forecasting model and in-situ observations capture individual solitons and
ISW packets (Figs. 6a and 6b). From 08 to 14 August, there is a notable increase in wave amplitude and nonlinearity,
reflecting the transition from neap to spring barotropic tides. The model's predictions show consistent arrival times,
baroclinic velocities (as indicated by color shades in Fig. 6), and maximum amplitudes (represented by contours in Fig. 6)
with those observed in the in-situ data. Although the model omits some small trailing waves (indicated by green arrows in
Fig. 6a) in the observations, it nonetheless demonstrates strong performance in forecasting ISWs at the initial 15 days.

Throughout the extended 15-day period, Figs. 6c and 6d continue to exhibit high consistency in predicting arrival times of
ISWs. However, the rates of false positives (simulated ISWs that are not observed, indicated by blue arrows in Fig. 6) and
false negatives (observed ISWs that are not reproduced, indicated by green arrows in Fig. 6) are relatively higher compared
to those observed during the initial 15 days. By considering all ISWs captured at the DS station in EXP. 1, the false positive
rate is 8.6% (3 out of 35) and the false negative rate is 11.4% (4 out of 35).

To assess the model's accuracy more quantitatively, we have identified the 28 well-predicted ISWs (indicated by red
arrows in Fig. 6). We extract their ISW characteristic parameters, such as arrival time, maximum vertical amplitudes,
baroclinic velocities, and wave direction, and compare them with field data (see red circles and green triangles in Fig. 7).
Note that detailed approaches to extract wave properties can be found in Gong et al. (2023). Firstly, the arrival times of ISWs
are displayed on the top and bottom of Fig. 7. The discrepancy between the model results and observations is consistently
less than one hour and a half, with a RMSD of 0.64 h, suggesting that the control run (EXP. 1) accurately captures arrival
times of ISWs. Secondly, the model's average of maximum vertical amplitude (~88 m) is comparable to the observed value
(~95 m), though the RMSD for this amplitude is 26.51 m. Thirdly, the average maximum baroclinic velocities are 1.34 m s$^{-1}$
in the model and 1.23 m s$^{-1}$ in the observations, respectively, with an RMSD of 0.39 m s$^{-1}$. Finally, the average wave





propagation directions are approximately 298° in the model and 288° in the observations, with an RMSD of 13.74°. Overall, EXP. 1 successfully reproduces the four key wave features of ISWs observed near the Dongsha Atoll.

## 4 Sensitivity experiments to evaluate model updates

Building on the standard experiment (EXP. 1), we modify its initial and boundary conditions, to individually assess the impact of turbulence parameter optimization, horizontally inhomogeneous stratifications and background currents on the prediction accuracy of ISW model in the NSCS. Details in configuration modifications are as follows (also see Table 1).

1) EXP. 2: Compared with EXP. 1, initial stratification profiles are horizontally homogeneous, derived from the seasonal-averaged WOA18 dataset. Moreover, background currents are excluded at both the initial conditions and boundary conditions.

2) EXP. 3: Compared with EXP. 2, mixing coefficients (i.e., viscosity and diffusivity) are imposed as $A_h = 0.5$ m$^2$ s$^{-1}$; $A_v = 5.0\times10^{-3}$ m$^2$ s$^{-1}$; $K_h = 0.5$ m$^2$ s$^{-1}$; $K_v = 5.0\times10^{-3}$ m$^2$ s$^{-1}$ in horizontal and vertical, respectively. Note that the setups in EXP. 3 are identical to those in ISWFM-NSCS v1.0 with the exception of the extended forecasting time (30 days).

3) EXP. 4: Compared with EXP. 1, background currents are only configured as initial conditions but no longer continuously imposed at four lateral boundaries.

### 4.1 Roles of optimized viscosity and diffusivity

Various 3D models with different configurations of viscosity and diffusivity have been previously employed to reproduce ISWs in the NSCS (e.g., Alford et al., 2015; Lai et al., 2019). Nonetheless, the determination of which turbulence coefficients are suitable to achieve the desired accuracy in ISW predictions remains unresolved. Here, we conduct two sensitivity experiments (EXP. 2 and EXP. 3) differing by two orders of magnitude in their viscosity and diffusivity coefficients.

Firstly, to assess the roles of turbulence parameters in stratification stabilities, we calculate the horizontally averaged buoyancy frequency profiles across the entire model domain and present these profiles over time in Fig. 8. It is evident that the stratification profiles in EXP. 3 gradually weaken over time, with a descending thermocline depth (see Fig. 8c). In contrast, the domain-averaged stratifications in EXP. 2, characterized by relatively smaller viscosity and diffusivity, remain significantly stable (see Fig. 8a). To better illustrate the crucial roles of optimized viscosity and diffusivity in maintaining stable stratifications, we present the averaged buoyancy frequency profiles at the beginning (black curves) and at the ending (red curves) of the model simulations, respectively (see Figs. 8b and 8d). In EXP. 2, the maximum buoyancy frequency in the thermocline slightly decreases from 0.018 s$^{-1}$ to 0.015 s$^{-1}$ after 30 days. Nonetheless, in EXP. 3, the maximum buoyancy frequency is nearly halved (from 0.018 s$^{-1}$ to 0.01 s$^{-1}$), rendering the model unable to accurately forecast ISW properties. Consequently, the valid forecast period in ISWFM-NSCS v1.0 is 15 days rather than 30 days.





Secondly, we present snapshots of sea surface height gradients and profiles in the primary propagation direction of ISWs from the first half and the second half of the simulation (12:00 UTC 11 August and 02:00 UTC 27 August, respectively) to compare horizontal characteristics and vertical structures of ISWs between EXP. 2 and EXP. 3 (Fig. 9). During the first 15 days (Figs. 9b and 9c), both EXP. 2 and EXP. 3 successfully reproduce westward-propagating ISWs, including both

individual solitons or ISW packets. However, during the extended 15 days (Figs. 9f and 9g), EXP. 2 and EXP. 3 tend to underestimate the ISW nonlinearities, consequently missing an ISW in the deep basin. Moreover, as the model runs for over 15 days, numerous spurious small eddies emerge in EXP. 3 (Fig. 9g) due to the accumulation of energy flux at the lateral boundaries.

Thirdly, to evaluate the sensitivity of the models in preproducing the ISW's vertical structures, a 7-day segment of

260 observational data (Fig. 10) was extracted from the DS station. This data includes temperature and velocities and spans the period from 25 August to 01 September 2014. Upon comparing Figs. 10b and 10c, it is evident that the characteristic half-widths observed in EXP. 2 are relatively narrower than those in EXP. 3, suggesting a higher degree of nonlinearity in ISWs with smaller turbulence coefficients over an extended period of 15 days. Furthermore, the stratification is much weaker in EXP. 3 than that in EXP. 2 (see contours in Figs. 10b and 10c). In addition, the statistical analysis, which includes data from

265 the initial 15 days (not shown), reveals that the rates of false positives and false negatives are 8.6% (3 out of 35) and 17.1% (6 out of 35) in EXP. 2, respectively. In contrast, in EXP. 3, the rates are 5.9% (2 out of 34) and 17.7% (6 out of 34), respectively, as indicated by the blue and green arrows.

Lastly, to quantitatively assess the performance of the sensitivity experiments with varying viscosity and diffusivity, we compare the biases of wave features for 28 ISWs (labelled by red arrows in Fig. 6) between EXP. 2, EXP. 3, and

270 observational data. In EXP. 2, the arrival time biases are generally under 1.5 h (blue triangles in Fig. 11a), with a RMSD of 0.77 h. In contrast, EXP. 3 shows a growing bias that exceeds 2 h over time (red triangles in Fig. 11a), resulting in a RMSD of 1.01 h. For maximum mode-1 amplitudes, both EXP. 2 and EXP. 3 tend to overestimate amplitudes during the first 15 days and underestimate them in the final 15 days (Fig. 11b), yielding RMSDs of 39.17 m and 40.39 m, respectively (Table 2). Regarding baroclinic velocities (Fig. 11c), RMSDs are 0.40 m s$^{-1}$ in EXP. 2 and 0.52 m s$^{-1}$ in EXP. 3. The RMSDs for

propagation directions are similar in both experiments, approximately 10.5° (Table 2). Lastly, EXP. 3 exhibits a significant discrepancy in modelling the ISW's characteristic half-widths, with a RMSD of 1.13 km, while EXP. 2 demonstrates better accuracy, with a RMSD of 0.28 km (Fig. 11e). Discrepancy mainly occurs towards the end of the forecast period, when stratification has notably weakened.

In summary, EXP. 3, utilizing the same turbulence configurations as ISWFM-NSCS v1.0, is able to practically predict the

280 key ISW features during the initial 15 days. However, the sensitivity model (EXP. 2), characterized by viscosity and diffusivity magnitudes two orders higher, presents a superior approach for identifying wave properties, particularly regarding wave nonlinearity, during the extended 15 days.



## 4.2 Roles of horizontally inhomogeneous stratification

The westward shoaling of thermocline, driven the northward-flowing Kuroshio, has been identified as a critical factor contributing to the west-east asymmetry of ISWs in the Luzon Strait (e.g., Zheng et al., 2007; Buijsman et al., 2010). However, the significant role of horizontally inhomogeneous stratification in this process has yet to be validated by a realistic ISW model. Here, we compare two sensitivity experiments (EXP. 2 and EXP. 4) with different initial conditions, namely EXP. 2 with horizontally homogeneous stratifications from WOA18 dataset and EXP. 4 with horizontally inhomogeneous stratifications from HYCOM reanalysis dataset.

First, we examine the effects of horizontally inhomogeneous stratifications on the spatial characteristics of ISWs and review Fig. 9. In EXP. 4, ISW crestlines are longer and more prone to distortion by background processes (Figs. 9d and 9h), which closely replicates the ISW scenario simulated in the control run (EXP. 1). Additionally, spurious eastward-propagating ISWs from the Luzon Strait appear in EXP. 2 (Fig. 9b), which are not reproduced in EXP. 4 (Fig. 9d). This discrepancy arises because the control run, initialized with the HYCOM reanalysis dataset, accounts for the east-west asymmetric thermocline associated with the Kuroshio. Moreover, the southern portion of ISW crestline is much more distinct in EXP. 4 (Fig. 9h) in comparison with EXP. 2 (Fig. 9f), especially as the ISWs approach the Dongsha Atoll and bifurcate into two branches.

Next, we analyse the differences in ISW vertical structures between EXP. 2 and EXP. 4 using data from the selected transect and the DS station. During the initial 15 days, successive westward-propagating internal solitons and ISW packets are captured along the transect both in EXP. 2 and EXP. 4 (Figs. 9b and 9d), as the stratification remain stable. In contrast, during the extended 15 days, EXP. 2 tends to underestimate the ISW nonlinearity, consequently missing an ISW in the deep basin (Fig. 9f), whereas EXP. 4 continues to reproduce it, albeit with a less significant amplitude (Fig. 9h). Given that the primary differences manifest during the final 15 days, we further compare the single-point outputs in EXP. 2 and EXP. 4. It is evident that EXP. 4 (Fig. 10d) captures more ISWs with narrower characteristic half-widths than EXP. 2 (Fig. 10b) at the DS station.

Finally, we conduct a quantitative assessment of the sensitivity models' ability to replicate ISWs by calculating the biases and RMSDs for five key wave properties (indicated by blue triangles and green stars in Fig. 11 and Table 2) in the cases with and without horizontally inhomogeneous stratification. Fig. 11a illustrates that bias of arrival time in EXP. 4 significantly exceed that in EXP. 2, resulting in a RMSD of 1.20 h in EXP. 4, compared to 0.77 h in EXP. 2. This may be due to the omission of the lateral boundary forcing in EXP. 4, resulting in the inability to continuously maintain horizontally inhomogeneous stratification. The RMSDs of maximum wave-induced velocities are very close (0.40 m s$^{-1}$ versus 0.44 m s$^{-1}$) in the two experiments (see Table 2). Nonetheless, EXP. 4 demonstrates superior performance in reproducing mode-1 wave amplitude with a RMSD of 31.94 m (versus 39.17 m in EXP. 2), as well as in accurately capturing propagation direction with a RMSD of 9.66$^{\circ}$ (versus 10.76$^{\circ}$ in EXP. 2).





In summary, horizontally inhomogeneous stratification is essential in the initial conditions of ISWFM-NSCS v2.0 particularly during the first 15 days. However, to sustain the west-east asymmetric stratification within the model domain, it is necessary to impose time-variable background currents at the four lateral boundaries.

### 4.3 Roles of background currents

        As inferred from Sec. 4.2, time-variable boundary conditions are crucial for maintaining the horizontally inhomogeneous
stratification within the model domain. Here, we extract the background currents (including temperature, salinity and velocities) from the HYCOM reanalysis dataset and impose them at the four lateral boundaries in the control run (EXP. 1). Then, we compare the model results from EXP. 1 with those from EXP. 4, which is solely initialized with 3D stratification and currents but does not include continuous lateral boundary forcing.

        Regarding the spatial distribution of ISWs, horizontal gradients of sea surface heights in EXP. 1 (Fig. 9a) exhibit a pattern
analogous to that observed in EXP. 4 (Fig. 9d) at 12:00 UTC on 11 August. However, a notable difference emerges at 02:00 UTC on 27 August, wherein the ISW crestlines in EXP. 1 are longer and more susceptible to distortion by background processes compared to those in EXP. 4. We subsequently examine the differences in the vertical structures of ISWs between two cases along the selected transect and over a 15-day time series at the DS station (Figs. 10a and 10d) during the extended 15 days. Although both cases successfully reproduce distinct vertical structures of ISWs along the transect, ISWs in EXP. 1
(Fig. 9e) exhibit greater nonlinearity compared to those in EXP. 4 (Fig. 9h), particularly within the deep basin. Fig. 10 illustrates that the rate of false positives is 8.6% (3 out of 35) both in EXP. 1 and EXP. 4, but the rate of false negatives (11.4%, 4 out of 35) in EXP. 1 is lower than that (17.1%, 6 out of 35) in EXP. 4.

        From a quantitative perspective, EXP. 1 demonstrates superior precision (47%) in predicting the arrival time of ISWs, as evidenced by a RMSD of 0.64 h, compared to a RMSD of 1.20 h in EXP. 4. This improved precision is attributed to the
presence of time-variable boundary conditions in EXP. 1, which results in a stable stratification. Conversely, in EXP. 4, the bias in arrival time progressively exceeds 1.5 h during the final 15 days (Fig. 11a). Additionally, the control run (EXP. 1) exhibits superior performance in reproducing maximum amplitudes, baroclinic velocities and half-widths than EXP. 4 (see black circles and green stars in Figs. 11b, 11c and 11e). Specifically, the RMSDs in EXP. 1 are 26.51 m, 0.39 m s$^{-1}$, and 0.17 km, whereas in EXP. 4 the RMSDs are 31.94 m, 0.44 m s$^{-1}$, and 0.50 km, respectively.

To sum up, by incorporating time-variable background currents at the lateral boundaries, the effects of background flows and mesoscale eddies on the propagation processes of ISWs are more accurately represented. This improvement enhances the model's accuracy in forecasting key wave features, such as arrival time, baroclinic velocities, maximum vertical amplitudes, and characteristic half-widths.





## 5 Conclusions

A robust 3D non-hydrostatic model ISWFM-NSCS v2.0 for forecasting ISWs in the NSCS has been presented. A reference test case was launched from 05 August 2014 and ran for 30 days, during which in-situ observations are available. Various wave properties are better characterized with ISWFM-NSCS v2.0 compared with ISWFM-NSCS v1.0. The major updates and findings are as follows.

    1) Optimized viscosity and diffusivity coefficients (i.e., $A_h = K_h = 1.0 \times 10^{-2}$ $m^2\,s^{-1}$, $A_v = K_v = 1.0 \times 10^{-5}$ $m^2\,s^{-1}$) contribute

to the stabilization of stratification profiles within the model domain, thereby extending the valid forecasting period to 30 days. By comparing the biases between sensitivity model results with in-situ observations at the DS station, we found that ISWFM-NSCS v1.0 gradually loses forecast precision regarding wave nonlinearities after 15 days. Specifically, the RMSD of ISW characteristic half-widths is 1.13 km in SWFM-NSCS v1.0 (EXP. 3), compared to 0.28 km with the optimized turbulence coefficients (EXP. 2).

2) Horizontally inhomogeneous stratifications are implemented as the initial conditions in ISWFM-NSCS v2.0, resulting in west-east asymmetric thermoclines on either side of the Luzon Strait. Considering these horizontally inhomogeneous stratifications, mode-1 wave amplitudes are more accurately reproduced, with a RMSD of 31.94 m in EXP. 4 versus 39.17 m in EXP. 2. Similarly, the propagation direction is better represented, with an RMSD of 9.66° in EXP. 4 versus 10.76° in EXP. 2.

3) Time-variable background currents at four lateral boundaries are essential for maintaining the horizontally inhomogeneous stratification within the model domain. Additionally, background circulations, such as the Kuroshio Current and mesoscale eddies, have been shown to significantly impact the behaviours and characteristics of ISWs in the NSCS. As compared with EXP. 4 (with RMSDs of 31.94 m, 0.44 $m\,s^{-1}$, and 0.50 km), applying the background currents could enhances the performance of ISWFM-NSCS v2.0 in reproducing maximum vertical amplitudes, baroclinic velocities, and

characteristic half-widths, resulting in improved RMSDs of 26.51 m, 0.39 $m\,s^{-1}$, and 0.17 km.

    In summary, ISWFM-NSCS v2.0 incorporates optimized turbulence coefficients, horizontally inhomogeneous stratifications, and background currents, compared with ISWFM-NSCS v1.0. As a result, ISWFM-NSCS v2.0 demonstrates considerable improvements in the model's ability to accurately predict a range of wave properties, achieving a 37% improvement in arrival time, a 34% improvement in mode-1 wave amplitude, a 25% improvement in wave-induced velocity,

and an 85% improvement in characteristic half-width.

*Code and data availability*. The MODIS satellite imagery can be freely downloaded from the NASA Worldview website (https://worldview.earthdata.nasa.gov, last access: 11 July 2024, Plato et al., 2019). The code of Massachusetts Institute of Technology general circulation model can be accessed at https://mitgcm.org/source-code/ (last access: 11 July 2024). The

input files, including initial and boundary conditions, as well as the corresponding output data for ISWFM-NSCS, are freely accessible through an open-access data repository available at https://doi.org/10.5281/zenodo.6792999.



*Author contributions*. YG, XC, ZC, QH, and SC developed the software. JX, RZ, and XZ provided guidance on the processing of observational data. YG conducted the numerical model. YG wrote the draft of the manuscript with the help of all the co-authors. All authors reviewed the final manuscript. YG, XC, ZC, and SC were responsible for conceptualisation, and for the workshop and training in the use of the software.

*Competing interests*. The contact author has declared that none of the authors has any competing interests.

*Financial support*. This work was jointly supported by the National Natural Science Foundation of China (NSFC) under contract Nos. 42130404, 42206012, 42276022, 42176025, 42276011 and U23A2032; Natural Science Foundation of Guangdong Province (2024A1515012703, 2024A1515012549); State Key Laboratory of Satellite Ocean Environment Dynamics, SIO, MNR (QNHX2308); the Science and Technology Projects of Guangzhou (2024A04J3587, 2024A04J9022); South China Sea Institute of Oceanology (SCSIO2023QY02 and LTOZZ2205). The numerical simulations were supported by the High Performance Computing Division, with assistance from HPC managers Wei Zhou and Dandan Sui at the South China Sea Institute of Oceanology.

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





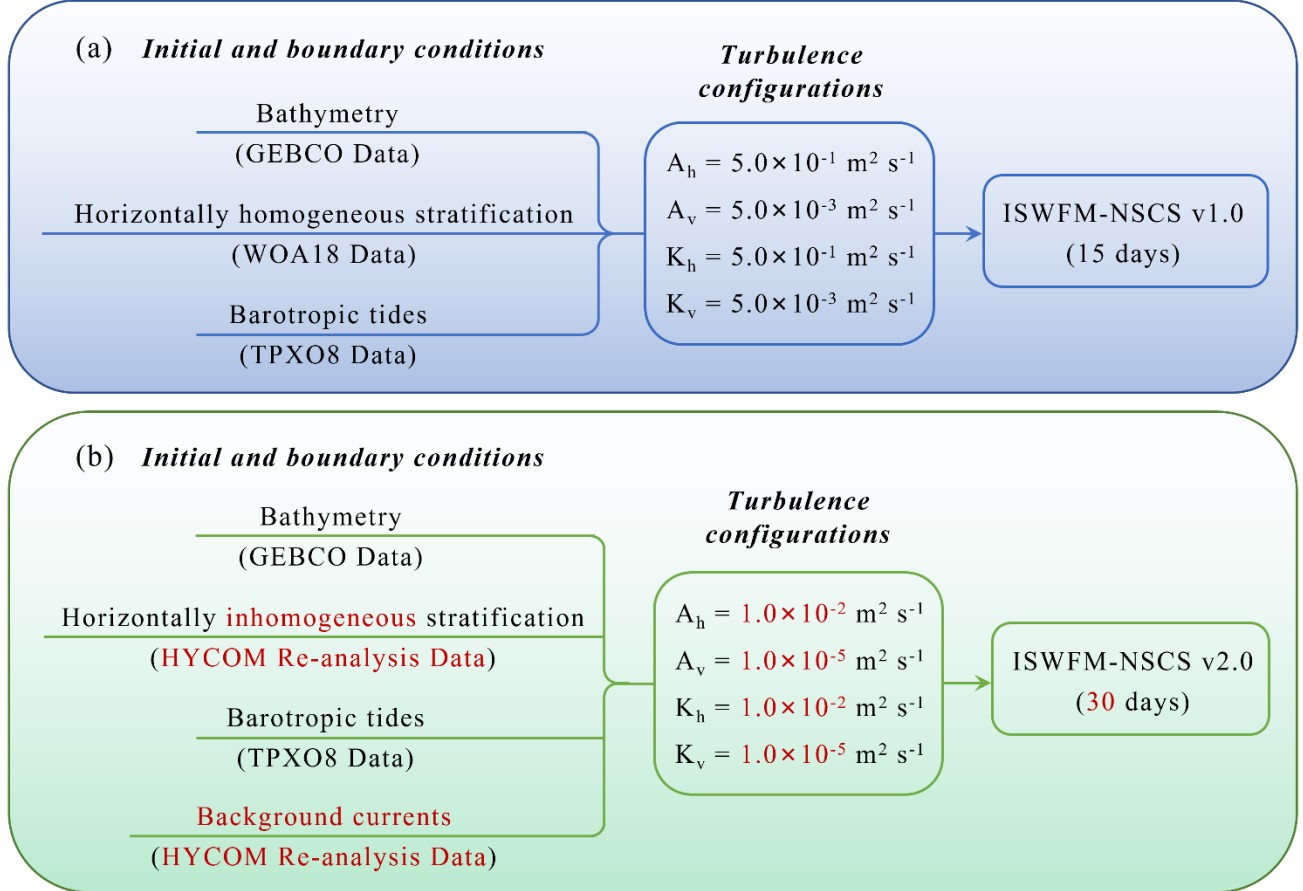

**Figure 1.** (a) The schematic of configuration and implementation of ISWFM-NSCS v1.0, which includes initial and boundary conditions. (b) Same as (a) but for ISWFM-NSCS v2.0. Note that the major model updates are marked in red in (b).





**Figure 2.** (a) Horizontally inhomogeneous temperature near the sea surface at the initial conditions (00:00 UTC 05 August 2014), which is derived from HYCOM reanalysis dataset. (b) Meridionally-averaged temperature profile through the entire model domain, showing west-eastward asymmetry of thermoclines. (c) Background velocity near the sea surface at the initial conditions, which is derived from HYCOM reanalysis dataset. (d) Model bathymetry, obtained by GEBCO dataset.




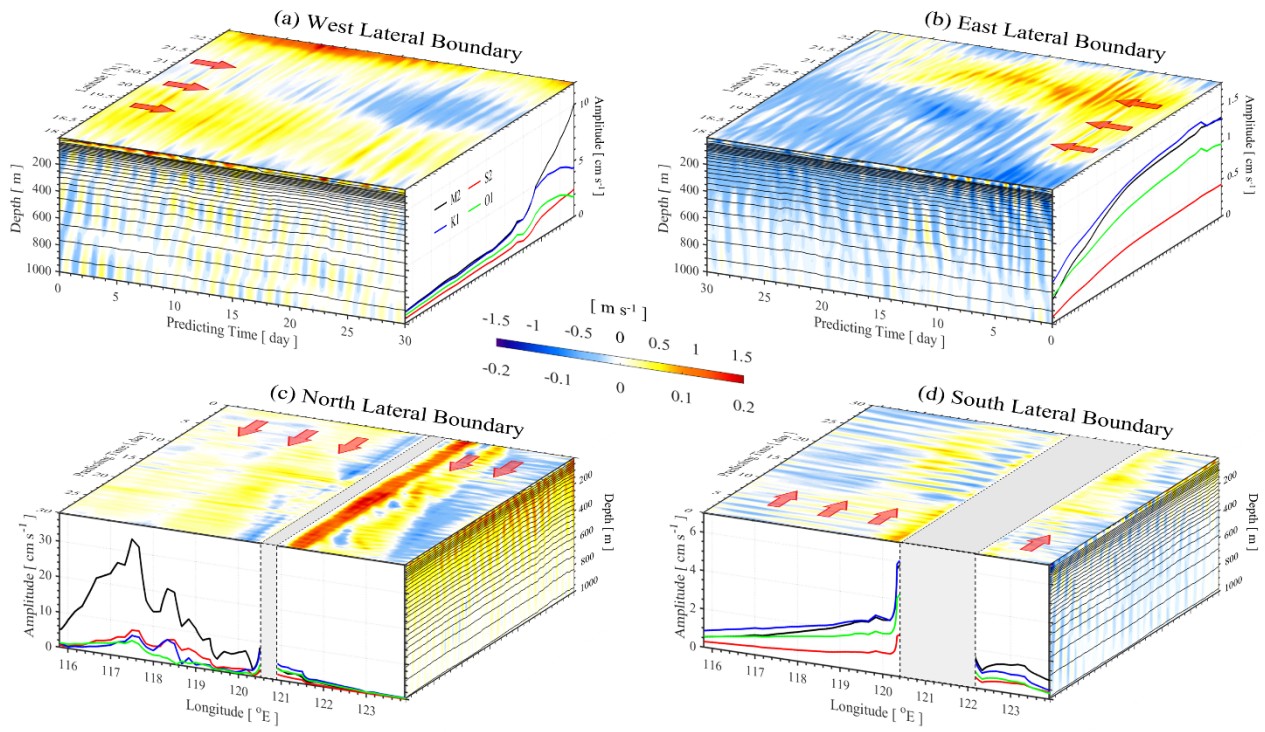

**Figure 3.** (a) Background currents and tidal forcing at the west lateral boundary, including x-t diagram of background zonal velocities at the sea surface (top panel), latitudinal averaged zonal velocity in the upper 1000 m (front panel), and tidal amplitudes of four primary constituents (right panel). Note that color ranges are -1.5 ~1.5 m s⁻¹ and -0.2 ~ 0.2 m s⁻¹ in the top

and front panels, respectively. (b-d) Same as (a) but for east, north and south lateral boundaries, respectively.



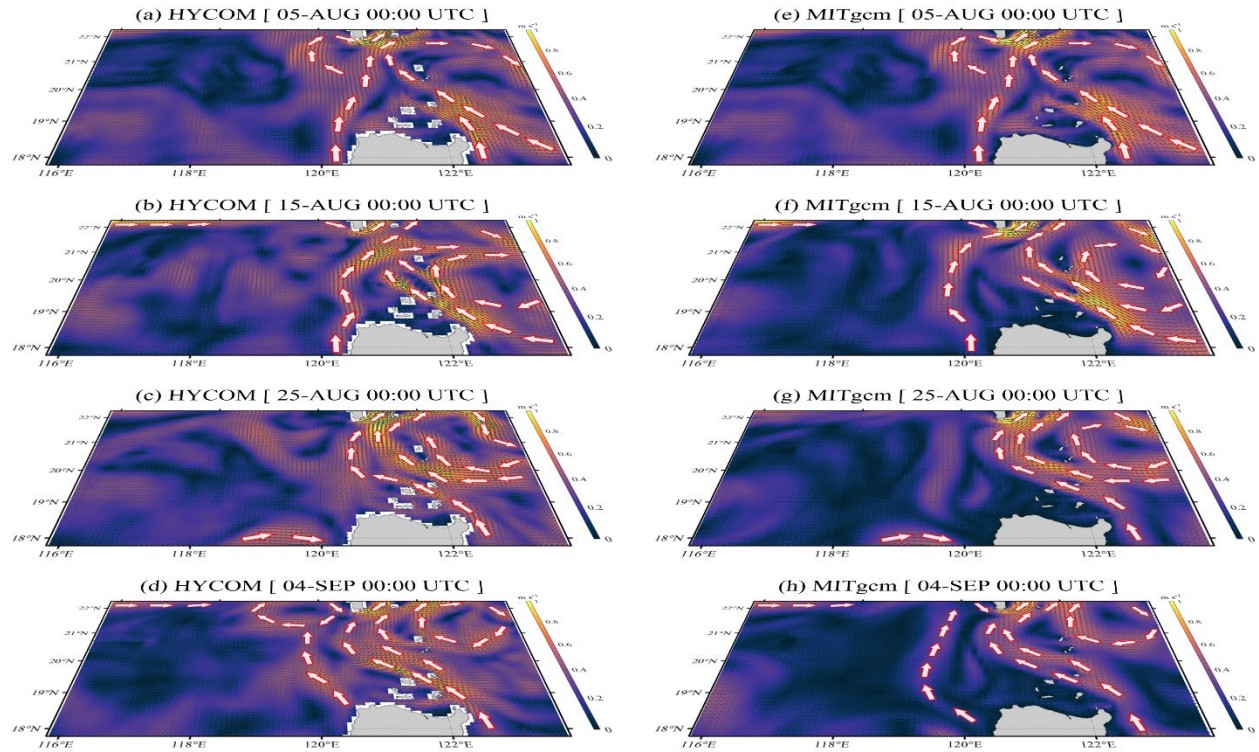

**Figure 4.** (a-d) Background currents and eddies at the sea surface from 05 August to 04 September in 2014 with a time interval of 10 days, derived from the global HYCOM reanalysis dataset (https://www.hycom.org/). (e-h) Same as (a-d) but derived from the model results of EXP. 0.





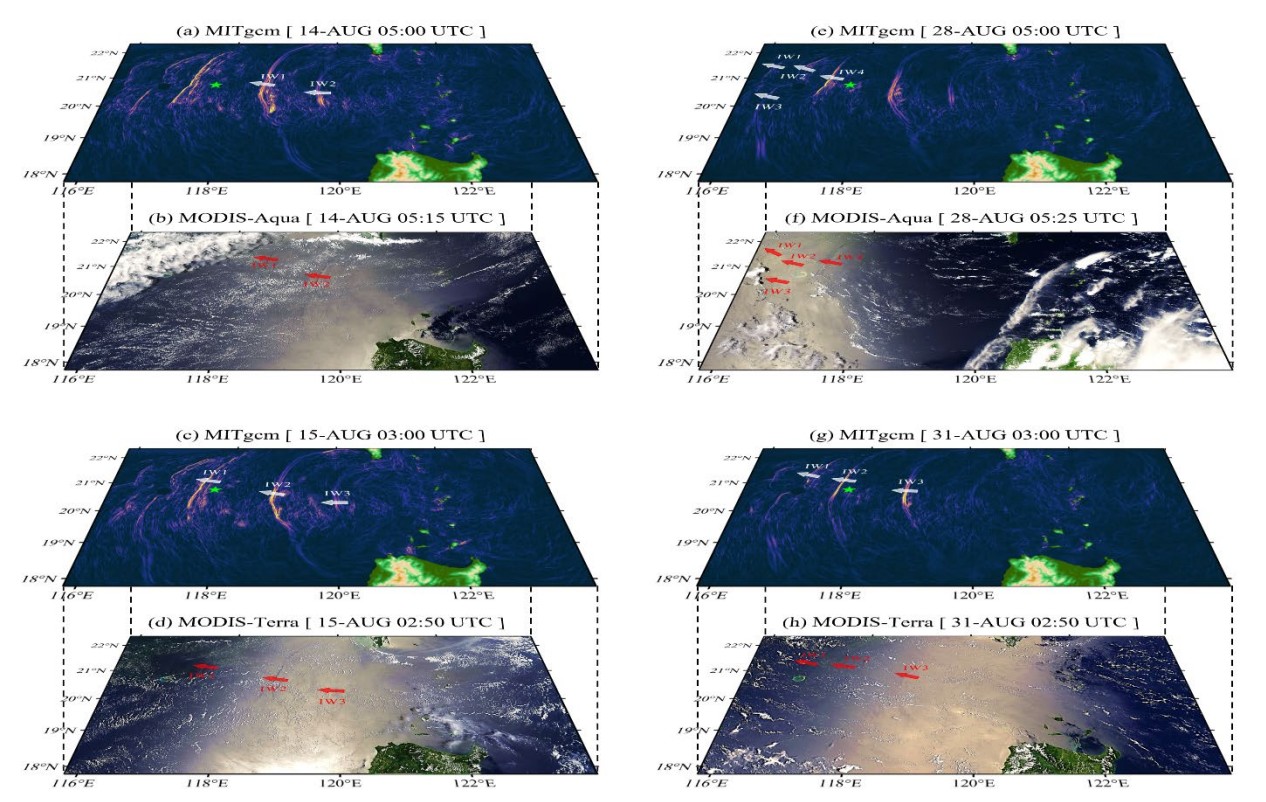

**Figure 5.** (a) Horizontal gradients of sea surface heights induced by internal solitary waves (ISWs) at 05:00 UTC on 14 August 2014, and (b) corresponding MODIS-Aqua image captured at 05:15 UTC on 14 August 2014. (c), (e), and (g) Same as (a) but at 03:00 UTC on 15 August, 05:00 UTC on 28 August, and 03:00 UTC on 31 August 2014, respectively. (d), (f), and (h) Same as (b) but with MODIS-Terra imagery at 02:50 UTC on 15 August, MODIS-Aqua imagery at 05:25 UTC on 28 August, and MODIS-Terra imagery at 02:50 UTC on 31 August 2014, respectively. Note that the MODIS images were freely accessible at the NASA Worldview website (https://worldview.earthdata.nasa.gov, open source).





**Figure 6.** (a) Temperature and wave-induced (or baroclinic) velocities along the main propagation direction of ISWs from 08 August to 15 August based on field observations at the DS station. (b) Same as (a) but for the standard experiment (EXP. 1). (c-d) Same as (a-b) but from 25 August to 31 September. Red arrows highlight ISWs correctly captured by the model, while blue and green arrows demote the false positive and false negative results, respectively.





**Figure 7.** Maximum mode-1 wave amplitudes (a), baroclinic velocities (b), and propagation directions (c) of twenty-eight ISWs at the DS station. Field observations are represented by red circles, while numerical model results are indicated by green triangles. Note that averaged values are shown as solid lines.



**Figure 8.** (a) Time series of horizontally-averaged background buoyancy frequency in the upper 500 m through the entire model domain in EXP. 2. (b) Black and red lines represent the averaged buoyancy frequency profiles at the beginning and at the ending of the model, respectively. (c-d) Same as (a-b) but in EXP. 3.







**Figure 9.** Sea surface height gradients, and temperature and baroclinic velocities along the transect (marked as dashed line in Fig. 8a) at 12:00 UTC on 11 August 2014 in the cases (a) EXP. 1, (b) EXP. 2, (c) EXP. 3, and (d) EXP. 4, respectively. (e-h) Same as (a-d) but at 02:00 UTC on 27 August 2014. Small panels in the bottom left display the zonal barotropic velocity (in m s$^{-1}$) in the Luzon Strait. Solid lines represent the barotropic tidal conditions at the specified times.





**Figure 10.** Time series of temperature and baroclinic velocities at station DS from 25 August to 01 September 2014 for the
model runs: (a) EXP. 1, (b) EXP. 2, (c) EXP. 3, and (d) EXP. 4. Red arrows highlight ISWs detected by the model, while
blue and green arrows denote the false positive and false negative results, respectively.



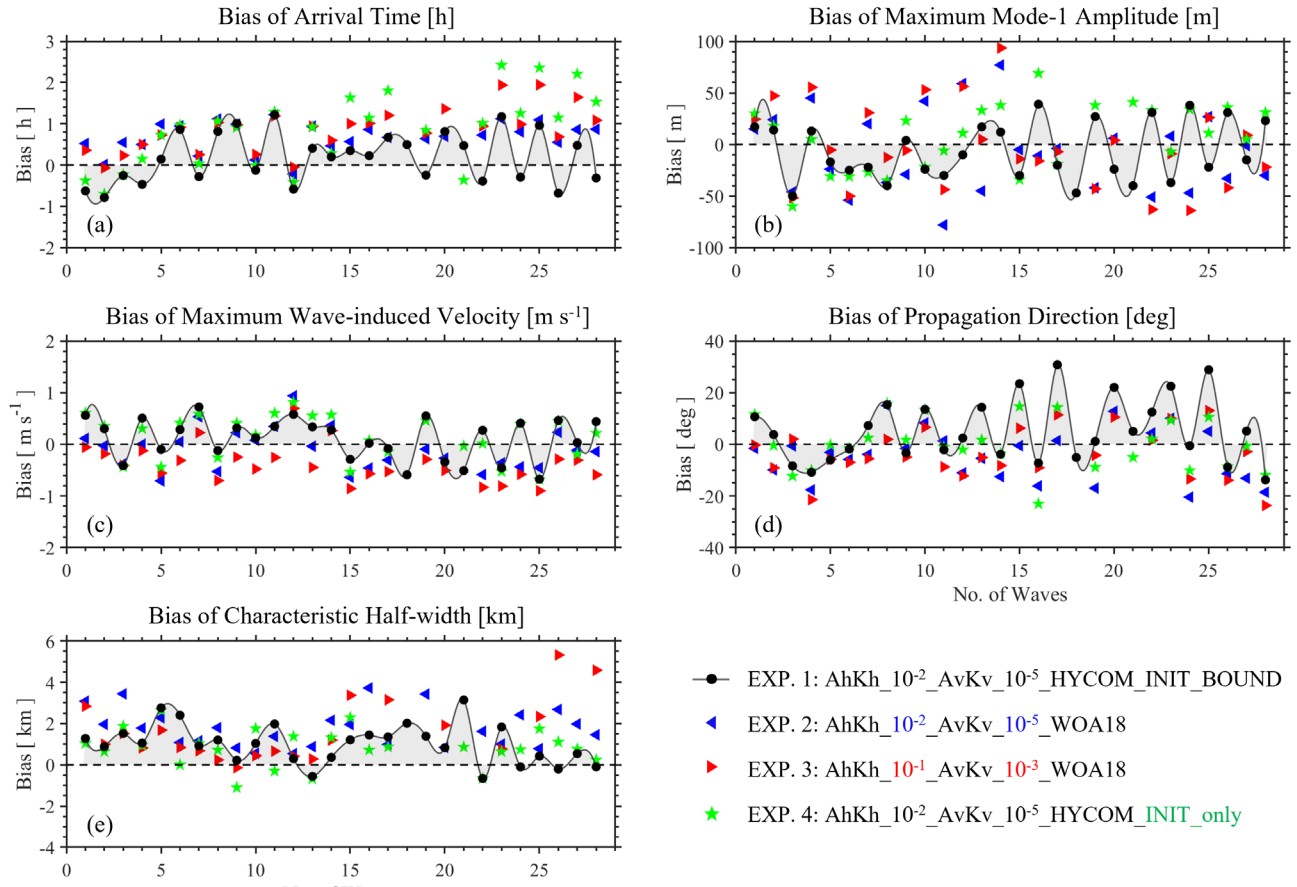

**Figure 11.** Bias in arrival time (a), maximum mode-1 amplitudes (b), wave-induced velocities (c), propagation directions (d), and half-widths (e) for twenty-eight ISWs at station DS. Different colours represent results from different experiments (EXP. 1 – EXP. 4).





**Table 1.** Summary of all experimental configurations.

| No. | $A_h, K_h$ | $A_v, K_v$ | Initial conditions | Boundary conditions |
|---|---|---|---|---|
| **EXP. 0** | $1.0\times10^{-2}$ m² s⁻¹ | $1.0\times10^{-5}$ m² s⁻¹ | 3D currents & Stratifications (HYCOM) | Background currents (HYCOM) |
| **EXP. 1** | $1.0\times10^{-2}$ m² s⁻¹ | $1.0\times10^{-5}$ m² s⁻¹ | 3D currents & Stratifications (HYCOM) | Surface tides (TPXO8) & Background currents (HYCOM) |
| **EXP. 2** | $1.0\times10^{-2}$ m² s⁻¹ | $1.0\times10^{-5}$ m² s⁻¹ | Horizontally homogeneous stratifications (WOA18) | Surface tides (TPXO8) |
| **EXP. 3** | $5.0\times10^{-1}$ m² s⁻¹ | $5.0\times10^{-3}$ m² s⁻¹ | Horizontally homogeneous stratifications (WOA18) | Surface tides (TPXO8) |
| **EXP. 4** | $1.0\times10^{-2}$ m² s⁻¹ | $1.0\times10^{-5}$ m² s⁻¹ | 3D currents & Stratifications (HYCOM) | Surface tides (TPXO8) |





**Table 2.** Root mean square deviation (RMSD) of five wave properties via comparing different sensitivity experiments with field observations at the DS station.

| No. | RMSD of arrival time [h] | RMSD of maximum mode-1 amplitude [m] | RMSD of baroclinic velocity [m s$^{-1}$] | RMSD of propagation direction [º] | RMSD of characteristic half-width [km] |
|---|---|---|---|---|---|
| **EXP. 1** | 0.64 | 26.51 | 0.39 | 13.74 | 0.17 |
| **EXP. 2** | 0.77 | 39.17 | 0.40 | 10.76 | 0.28 |
| **EXP. 3** | 1.01 | 40.39 | 0.52 | 10.09 | 1.13 |
| **EXP. 4** | 1.20 | 31.94 | 0.44 | 9.66 | 0.50 |