# Peer review of "ISWNM-NSCS v2.0: advancing the internal solitary wave numerical model with background currents and horizontally inhomogeneous stratifications"

_Geoscientific Model Development, 2024_

## Author Comment (AC1)

**Response letter to the Chief-Editor-Comment**

You have not published a full version of the ISWFM-NSCS v2.0 model. This is forbidden by our policy, and your manuscript should not have been accepted for Discussions because of it. All the code must be published openly and freely to anyone before submission of a manuscript. Also, you have not published a version of MITgcm in a permanent repository that we can accept.

Therefore, we are granting you a short time to solve this situation. You have to reply to this comment in a prompt manner with the information for the repositories containing all the models, and code that you use to produce and replicate your manuscript. The reply must include the link and permanent identifier (e.g. DOI). Also, any future version of your manuscript must include the modified section with the new information.

***Response*:**

We appreciate your comments on our code repository. According to your suggestion, we have read the code and data policy of GMD in detail, and have added all model code on the Zenodo repository with the DOI number https://doi.org/10.5281/zenodo.14841770.

Therefore, the "Code and data availability" is now rewritten as "The MODIS satellite imagery can be freely downloaded from the NASA Worldview website (https://worldview.earthdata.nasa.gov, last access: 11 July 2024, Plato et al., 2019). The input files, including initial and boundary conditions, as well as the corresponding output data for ISWFM-NSCS v2.0, are freely accessible through an open-access data repository available at https://doi.org/10.5281/zenodo.14841770 (Gong, 2025, last access: 10 February 2025)".

---

## Author Comment (AC2)

**Response letter to the Chief-Editor-Comment**

We appreciate your willingness to comply with the policy; however, it is my interpretation that the repository that you have linked in your response is not enough to comply with the policy, as it does not contain all the code necessary to replicate your work, but only a few files to run with the main model. It is your responsibility to provide all the code that you use and is necessary.

Also, I strongly recommend you to provide separated repositories for the code and for the data. Currently it is necessary to download the output files to check the code, which is not operational.

*Response*:

Thank you for your valuable suggestions. We appreciate your concerns regarding the completeness and operability of the code, so we now provide further clarification on this matter below.

Our model is built on the MITgcm (MIT General Circulation Model) framework, and its operation depends on the MITgcm environment. MITgcm is an open-source ocean model, and its source code and documentation are publicly accessible on its official website (https://mitgcm.org/source-code/, last access: 10 February 2025). Our research extends and optimizes the MITgcm framework, and as such, the model requires the MITgcm environment to function properly—a point explicitly highlighted in the repository description. In the submitted code repository, we have included the core code files necessary for interfacing with MITgcm, which are critical for running our model. However, I am uncertain whether creating a Zenodo repository for the MITgcm source code might violate its licensing terms or intellectual property rights. Could you please clarify this?

Additionally, we fully agree with your suggestion to separate the code and data. In the updates, we now store the code and data in separate files and provide detailed instructions to make it easier for users to access and run our model (https://doi.org/10.5281/zenodo.14842090).

Overall, the "Code and data availability" is now rewritten as "The MODIS satellite imagery can be freely downloaded from the NASA Worldview website (https://worldview.earthdata.nasa.gov, last access: 11 July 2024, Plato et al., 2019). The code of Massachusetts Institute of Technology general circulation model can be accessed at https://mitgcm.org/source-code/ (last access: 10 February 2025). The input files, including initial and boundary conditions, as well as the corresponding output data for ISWFM-NSCS v2.0, are freely accessible through an open-access data repository available at https://doi.org/10.5281/zenodo.14842090 (Gong, 2025, last access: 10 February 2025)".

---

## Author Comment (AC3)

**Response letter to the Chief-Editor-Comment**

The MITgcm license (https://github.com/MITgcm/MITgcm/blob/master/LICENSE.txt), which you should have read and know, as you are using the model and one would expect that you know what you can do or not with it, clearly states that you can redistribute the code. Therefore, you can and must store the MITgcm version that works with your developed code in a permanent repository, and provide its details, link and permanent handle (e.g. DOI). In this regard, in the manuscript, you should indicate with what MITgcm version is compatible your code (it could happen that it is not compatible with a future version), and in the Code Availability section, instead of the MITgcm webpage that you currently mention, you should include the new repository that you set up with the frozen version of the code that you use here.

***Response*:**

We appreciate your comments on our code repository. According to your suggestion, we have read the MITgcm license in detail, and have added required MITgcm code on the Zenodo repository with the DOI number https://doi.org/10.5281/zenodo.14847454.

Therefore, the "Code and data availability" is now rewritten as "The MODIS satellite imagery can be freely downloaded from the NASA Worldview website (https://worldview.earthdata.nasa.gov, last access: 11 July 2024, Plato et al., 2019). The code of Massachusetts Institute of Technology general circulation model for ISWFM-NSCS v2.0 can be accessed at https://doi.org/10.5281/zenodo.14847454 (last access: 11 February 2025). The input files, including initial and boundary conditions, as well as the corresponding output data for ISWFM-NSCS v2.0, are freely accessible through an open-access data repository available at https://doi.org/10.5281/zenodo.14842090 (Gong, 2025, last access: 10 February 2025)".

---

## Author Comment (AC4)

**Response letter to the referee #1**

Presented is an improved version of the internal solitary wave model. The efforts of the improvements have been made through several aspects: inclusion of background currents, horizontally inhomogeneous stratifications, and many others. The results are very encouraging. This paper documents these achievements towards a better internal solitary wave model. The graphs are impressively beautiful. I would recommend it be accepted for publication after some clarifications and minor revision.

**Response:**

We would like to thank the referee for the careful reading and valuable comments. In the revision, we have carefully considered them, and the necessary changes are provided to address them. Below, we provided point-by-point responses in blue to your comments.

Specific comments:

The title gives readers an impression that this is a forecast model. However, it has not been implemented as a forecast model yet. The numerical experiments reported in this paper are hindcast modeling. I would suggest to remove the keyword "forecasting".

**Response:** We sincerely appreciate the referee's insightful suggestion. In accordance with this recommendation, we have systematically replaced the abbreviation "ISWFM" (Internal Solitary Wave Forecasting Model) with "ISWNM" (Internal Solitary Wave Numerical Model) throughout the manuscript. This revision ensures consistency with the technical scope of our model, which emphasizes numerical simulation capabilities rather than operational forecasting. All relevant figures and texts have been updated to reflect this terminology adjustment.

L17, "ISWFM-NSCS v2.0" can be replaced with "the new version".

**Response:** "ISWFM-NSCS v2.0" has now been replaced with "ISWNM-NSCS v2.0".

L20, "in ISWFM-NSCS v2.0" can be removed.

**Response:** "in ISWFM-NSCS v2.0" is now removed.

L21, "presented" should be changed to "used".

**Response:** We now replace "presented" with "used".

L68, "by comparing with numbers of" can be changed to "through".

*Response*: We now replace "by comparing with numbers of" with "through".

L79, "by" should be changed to "by using" or "from", and "a" should be "the".

*Response*: We now replace "by a" with "by using the".

L87, "well" should be changed to "good".

*Response*: We now replace "well" with "good".

L138-139, it would be good to provide some examples, e.g., Liu et al. (2000, 2008).

*Response*: We fully concur with the referee's suggestion regarding the importance of exemplifying the interactions between ocean circulations and internal solitary waves. In revised manuscript, we have now integrated two pivotal studies by Liu et al. (2000, 2008) here.

Figure 5, the model name "MITgcm" should be changed to "ISWFM-NSCS v2.0" for a more specific case.

*Response*: We agree that the headings should be more specific and now replace "MITgcm" with "ISWFM-NSCS" in Figure 5.

L145-148, it is mentioned that background currents are added and that velocity data are also extracted from the HYCOM. However, it is not clear whether the HYCOM velocity data are used as the background currents. If so, please clearly state it in this paragraph. If not, then provide more information on what data are used for specifying the background currents or temperature and salinity fields in the ISWFM-NSCS v2.0. (This information is found later in the manuscript, L320-321. But it would be better to state that earlier in this paragraph).

*Response*: We sincerely agree that the information about how to extract HYCOM velocity and associated temperature and salinity fields as the initial and boundary conditions should be clarified in section 2.3. To demonstrate the source of background currents and hydrographic fields in the ISWNM-NSCS v2.0, we have now explicitly stated in the revised manuscript as follows:

"The background zonal and meridional velocity fields, associated with the corresponding temperature and salinity fields, are directly derived from the global HYCOM re-analysis dataset (https://www.hycom.org/, last access: 12 July 2024). These three-dimensional datasets are linearly

interpolated onto the model grid to initialize the baseline dynamic conditions, while the time-varying velocity fields from the HYCOM dataset are imposed as lateral boundary forcing across all four domain edges, thereby continuously driving the internal circulation patterns through dynamic coupling."

---

## Author Comment (AC5)

**Response letter to the referee #2**

This paper presents an evolution of an already existing model focused on the internal solitary wave forecasting, known to be challenges for many physical reasons. Based on my own reading, the paper is clear, well written and present several important results that might interest the community. Three major evolutions are presented: modification of the eddy viscosity/diffusivity, add of a realistic non-homogeneous stratification and background current. The authors showed the impact of adding each component on the forecasting performances.

**Response:**

We would like to thank the referee for the careful reading and valuable comments. In the revision, we have carefully considered them, and the necessary changes are provided to address them. Below, we provided point-by-point responses in blue to your comments.

However, my main concern is about the choice of a constant eddy viscosity/diffusivity for their model. If the value is well suited for ISW as the authors explained, the values are far too large out of these phenomena, probably rapidly destroying the stratification, a problem they already faced in the first version of their model. I would suggest using an adaptative turbulent closure scheme, like the classical k-epsilon. At least, I would appreciate such a test or a discussion on that.

**Response:**

We sincerely appreciate the reviewer's insightful comment regarding the application of constant eddy viscosity/diffusivity and the suggestion to explore adaptive turbulent closure schemes. To address this concern, we conducted a sensitivity experiment using the K-Profile Parameterization (KPP) scheme (Large et al., 1994), a widely adopted turbulent closure model for oceanic boundary layers. Simulations incorporating the KPP scheme improved arrival time predictions (RMSD = 0.58 h vs. 0.63 h), but underperformed the control run (constant coefficients) in reproducing critical ISW properties, including maximum amplitude (RMSD = 37.22 m vs. 26.51 m), propagation direction (RMSD = 14.46° vs. 13.74°), and half-widths (RMSD = 0.25 km vs. 0.17 km). These discrepancies align with findings by Thakur et al. (2022), where the KPP scheme's excessive vertical mixing in stratified regions dampened ISW signals and degraded wave coherence. Statistical results and comparative metrics are detailed in Appendix A (Fig. A1). Given these findings, we retain the constant-coefficient configuration as the control run and relegate the KPP analysis to Appendix A to maintain focus on the optimized model setup.

Another important remark regarding the choice of the eddy viscosity/diffusivity, concerns the

numerical diffusivity which is neither taken into account nor discussed. I would appreciate the author to warn on this potential issue, when they explain the choice of the values based on in situ measurements.

**Response:**

We sincerely appreciate the reviewer's insightful comment regarding numerical diffusivity and its potential influence on our model's parameterization. Below, we have revised the relevant text to explicitly address this concern, while maintaining consistency with observational and theoretical justifications for our parameter choices. The added paragraph in section 2.1 emphasizes the distinction between physical and numerical diffusivity and acknowledges the latter as a factor requiring consideration. Specific revisions are detailed below:

"Numerical diffusivity, an artifact of discretization in the model's advection schemes, may also contribute to the effective mixing in simulations. While our parameter choices are grounded in observational constraints, the total diffusivity experienced in the model could include both physical and numerical components. To mitigate this, ISWNM-NSCS v2.0 employs a third-order nonlinear advection scheme to minimize spurious numerical diffusion (Adcroft et al., 2008). Nevertheless, future work will aim to explicitly quantify and disentangle these effects, as numerical diffusivity remains an important consideration in interpreting model-derived mixing rates."

**Minor suggestions:**

The introduction is lacking details on the amplitude/wavelength of the ISW in the region. It's important to be fully convince that you resolution is enough to really capture these small-scale waves. In addition, I fell that it is lacking geographical information about the region. Maybe you can add the quoted names on Fig 2.

**Response:** We thank the reviewer for the valuable feedback. To address the concerns, we have added observed ISW characteristics in the NSCS (vertical displacements exceeding 200 m and wavelengths of  $\sim$ 3 km) to emphasize the necessity of high-resolution modelling. ISWFM-NSCS v1.0's horizontal (500 m) and vertical (90 layers) resolution is explicitly stated to align with these scales. Geographic clarity is enhanced in Fig. 2 with labelled features (e.g., Dongsha Atoll, Luzon Island, Taiwan Island, SCS, and West Pacific) and bathymetric contours.

Line 155 it is not clear if you are talking about 28 properties or 28 ISWs.

*Response*: We fully agree that the original phrasing was unclear and have revised it to "five wave properties of 28 ISWs" to improve clarity.

Regarding the observations, even if they are already introduced in Gong et al 2023, I suggest to briefly introduce them: which instrument, how long ...

*Response*: We entirely agree that a brief introduction of in-situ observation should be added and now revise the sentence as "To conduct a more detailed evaluation of the model's accuracy in predicting ISWs, we incorporate field observations from the Dongsha (hereafter DS) mooring station (117°44.7'E, 20°44.2'N; deployed from 1 August to 6 September, 2014). The mooring included ADCPs (2-min sampling; 16/8-m vertical bins) and distributed temperature, CTD, and CT sensors (10–15 sec sampling). More details can be found in Gong et al. (2023)."

In section 4, test the statistical significance of your differences. Especially in lines 310.

**Response:** Thank you for highlighting the need for statistical rigor. We have now conducted independent two-sample t-tests (assuming unequal variances) to evaluate the significance of differences between in-situ observational wave properties and those in the numerical experiments (EXP. 1 - EXP. 5) in the revised manuscript.

Fig 8. Please add the legend in the panels b and d. *Response:* We have now added the legend in panels (b) and (d).

Fig 11: It is quite hard reading this figure. It would suggest to move to a table that summarize the averaged values should be much more readable. In particular, I recommend to separate the 15 first days and the 15 last days for your analysis.

*Response*: We fully agree with the reviewer that the original Fig. 11 was a bit hard to read. The figure has been restructured into two columns: the left column displays biases for five ISW properties during the initial 15-day period, while the right column corresponds to the final 15 days. Additionally, we expanded Table 2 to summarize the root-mean-square deviations (RMSD) of these properties across both periods, enabling a clearer comparative analysis. Corresponding text revisions have been made to align with these updates. Thank you for this constructive feedback.

---

## Referee Report (RR1)

This manuscript presents a realization of the MIT gcm as a regional modeling tool.  It is admirable to try to tackle a large geographical region such as the northern South China Sea.  The model builds on an earlier iteration of modeling and thus I presume is already being used in one form or another.

I have used the MITgcm myself, and understand its utility in this context.  My usual toolbox is more process study oriented and is typically high order.  I am keenly aware that this cannot be the choice for the present authors.  Nevertheless, the modelling presented in this manuscript requires more clarity.

The resolution is likely controlled by the need to resolve such a large area, but if the aim is to represent internal solitary-like waves, the large grid spacing needs justification.  The reader should know how many points per wave a typical wave form, and how this compares to the standard in process studies.  The resolution seems really low to me, but then I am one of those process study modellers I mentioned in my previous sentence.

Similarly the discussion of turbulence is a bit misleading.  The authors quote a number of turbulence schemes applied at different scales and use this to justify constant eddy viscosity values in the vertical and horizontal.  These values strike me as at least an order too high to me.  The reader needs to know how a single ISW would be affected by these choices.  Such model runs should take a day or

so, and their results can be summarized in a table.  These would provide an important counterpoint to the rather ambiguous validations provided in the present version.

While not a theory paper, what is presented on ISWs is pretty dodgy.  Gear and Grimshaw is a very old paper, the results of which have been superseded by other (often open source) tools.  There are even monographs on the theory which would provide a more modern link to discussion, literature and codes.

The discussion needs to be cleaned up as some parts read very strangely.  There are various tools the authors can use for this.  In terms of content, the aforementioned turbulence models are presented as interchangeable when as a point of fact they are designed for very different things (i.e. Mellor-Yamada versus Gent-McWilliams).  There are also strange statements about the source of turbulence that seem at odds with my understanding of ocean physics.

Finally, I was left wondering how the present methodology compares and contrasts with well established models like Getm-Gotm when applied to something like the Baltic Sea.  The two tools are different in purpose, but it would help the context to contrast them.

I realize this is a rereview, but this is my first time seeing this manuscript, and it needs significant revision before it can be deemed "ready to publish".

---

## Author Response (AR2)

**Response letter to the referee #3**

This manuscript presents a realization of the MITgcm as a regional modeling tool. It is admirable to try to tackle a large geographical region such as the northern South China Sea. The model builds on an earlier iteration of modeling and thus I presume is already being used in one form or another.

I have used the MITgcm myself, and understand its utility in this context. My usual toolbox is more process study oriented and is typically high order. I am keenly aware that this cannot be the choice for the present authors. Nevertheless, the modelling presented in this manuscript requires more clarity.

*Response*:

We would like to thank the referee for the careful reading and valuable comments. We entirely agree that the modelling presented in this manuscript requires more clarity. Therefore, in the revision, we have carefully considered them and the necessary changes are provided to address them. Below, we provided point-by-point responses in blue to your comments.

The resolution is likely controlled by the need to resolve such a large area, but if the aim is to represent internal solitary-like waves, the large grid spacing needs justification. The reader should know how many points per wave a typical wave form, and how this compares to the standard in process studies. The resolution seems really low to me, but then I am one of those process study modelers I mentioned in my previous sentence.

*Response*:

We sincerely thank the reviewer for raising this valuable comment regarding horizontal resolution and its implications for resolving internal solitary waves (ISWs). We fully agree that justifying grid spacing is essential for process-oriented studies, and we appreciate the opportunity to clarify this aspect.

In the original submission, the discussion of resolution sensitivity was omitted because the ISWFM-NSCS v1.0 framework (Gong et al., 2023, GMD) already evaluated the impacts of horizontal resolutions ($\Delta x$ = 250 m, 500 m, and 1000 m) on internal wave reproductions. Key findings from that study are listed below:

(1) $\Delta x$ =1000 m: Failed to accurately resolve key features of ISWs in the northern South China Sea (NSCS), such as waveform, maximum amplitude and characteristic half-width.

(2) $\Delta x$ = 500 m: Captured the fundamental characteristics of ISWs (e.g., waveform, maximum amplitude, and wave-induced velocity) with reasonable precision, achieving approximately 6–8 grid points per characteristic half-width for typical ISWs in the SCS. This aligns well with standard practices in regional ISW modeling studies (e.g., Zhang et al., 2011; Lai et al., 20219), which often

employ 4–10 points per characteristic half-width depending on nonlinearity and nonhydrostatic effects.

(3) $\Delta x = 250$ m: Improved the accuracy of wave properties (e.g., a ~40% reduction in characteristic half-width bias compared to the $\Delta x = 500$ m case). However, this refinement increased computational costs by a factor of 5 for the same domain, posing significant challenges for operational forecasting timeliness.

In the current version (ISWNM-NSCS v2.0), our focus lies in optimizing initial/boundary conditions and turbulence configurations rather than further refining resolution. Nevertheless, we recognize the importance of contextualizing our chosen 500 m resolution. Accordingly, in the revised manuscript (Section 2, Lines 81–86), we have now explicitly stated the grid points per wavelength (6–8) for typical ISWs in the NSCS and compared this to established resolution standards in process studies (e.g., Zhang et al., 2011; Lai et al., 2019). Moreover, we now highlight the trade-off between computational efficiency and precision, emphasizing that 500 m strikes a balance for regional-scale forecasting while retaining dynamical precision, and reference the last version (Gong et al., 2023) for readers seeking detailed resolution sensitivity analyses.

We acknowledge that higher resolutions (e.g., 250 m) would benefit small-scale process studies, but our priority here is to advance the model's operational readiness for the SCS basin. In future work, it would be interesting to further explore adaptive mesh refinement to locally enhance resolution in critical areas (e.g., wave generation sites) without inflating unnecessary computational costs.

Similarly, the discussion of turbulence is a bit misleading. The authors quote a number of turbulence schemes applied at different scales and use this to justify constant eddy viscosity values in the vertical and horizontal. These values strike me as at least an order too high to me. The reader needs to know how a single ISW would be affected by these choices. Such model runs should take a day or so, and their results can be summarized in a table. These would provide an important counterpoint to the rather ambiguous validations provided in the present version.

***Response***:

We sincerely appreciate the reviewer's insightful critique regarding our treatment of turbulence parameterization and the justification for constant eddy viscosity values. We acknowledge that further quantification of viscosity and diffusivity impacts on ISW strengthens model transparency, and we have conducted comprehensive sensitivity tests (EXPs. A1-A8, K1-K8) spanning four orders of magnitude (0.01× to 100× CTRL values from EXP. 1) to address this point.

In the EXPs. A1-A4, the horizontal viscosity coefficient ($A_h$) is ranging from 100×CTRL (1.0×

$10^0$ m$^2$ s$^{-1}$) to 0.01×CTRL (1.0×10$^{-4}$ m$^2$ s$^{-1}$). As a result, the EXPs. A1-A4 reveal exceptional stability in four key ISW properties across different scaling factors. Specifically, the arrival time (0.64 h), baroclinic velocity (~0.40 m s$^{-1}$), and propagation direction (13.67°–13.74°) show negligible RMSD variations in EXP. A1–A4. Only maximum mode-1 amplitude exhibits mild degradation (i.e., 27.53–28.59 m vs. 26.51 m in CTRL), while characteristic half-width shows marginal improvement in EXP. A2 (i.e., 0.15 km vs. 0.17 km in CTRL). Conversely, the EXPs. A5-A8 show greater sensitivity for vertical eddy viscosity. In details, EXP. A5 significantly degrades propagation direction (16.05° vs. 13.74° in CTRL), and most sensitivity experiments for $A_v$ worse reproduce maximum wave amplitudes. Although EXP. A6 slightly improves arrival time (0.58 h vs. 0.64 h in CTRL), no sensitivity experiments achieve >5% improvement across multiple ISW properties.

However, in the EXPs. K1-K4, modifications to horizontal diffusivity ($K_h$) yield mixed results. While arrival time and baroclinic velocity remain stable (±0.04 h and ±0.02 m s$^{-1}$), EXP. K3 ($K_h$=0.1×CTRL) substantially degrades maximum wave amplitude predictions (31.85 m vs. 26.51 m in CTRL). Characteristic half-width consistently worsens (0.18–0.22 km vs. 0.17 km in CTRL), though EXP. K4 slightly improves prediction of propagation direction (13.16° vs. 13.74° in CTRL). For vertical diffusivity (EXPs. K5-K8), extreme scaling causes pronounced effects. Specifically, although EXP. K5 ($K_v$=100×CTRL) shows a slight improvement in arrival time (0.55 h vs. 0.64 in CTRL), it significantly degrades four other wave properties simultaneously, namely maximum amplitude (31.92 m vs. 26.51 m in CTRL), baroclinic velocity (0.45 m s$^{-1}$ vs. 0.39 m s$^{-1}$ in CTRL), propagation direction (15.26° vs. 13.74° in CTRL), and characteristic half-width (0.28 km vs. 0.17 km in CTRL). Conversely, EXP. K7 ($K_v$=0.1×CTRL) improves maximum amplitude (27.10 m vs. 26.51 m in CTRL) and baroclinic velocity (0.38 m s$^{-1}$ vs. 0.39 m s$^{-1}$ in CTRL), but this is offset by half-width degradation (0.21 km vs. 0.17 km in CTRL).

Overall, no sensitivity experiment outperforms CTRL across all five ISW properties, but only isolated cases (e.g., arrival time in the EXP. A6) show >5% improvement in single metrics. The CTRL run maintains the most balanced performance, with all RMSDs within intermediate ranges. The fluctuations in ISW properties across all 16 sensitivity experiments confirm that viscosity and diffusivity configurations appear robust in the CTRL run (EXP. 1).

While not a theory paper, what is presented on ISWs is pretty dodgy. Gear and Grimshaw is a very old paper, the results of which have been superseded by other (often open source) tools. There are even monographs on the theory which would provide a more modern link to discussion, literature and codes.

***Response*:**

We sincerely thank the reviewer for this valuable comment regarding the theoretical foundation

of our ISW discussion. We fully acknowledge that the original citation of Gear and Grimshaw (1983) represented an outdated reference point that failed to reflect contemporary advances in solitary wave theory. In this revision, we have carefully restructured the fundamental introduction and theoretical framework throughout the manuscript, eliminating obsolete citations and integrating modern developments. The revised text now builds upon seminal treatments from Grimshaw et al. (2010) for extended KdV theory and Simmons et al. (2010) for non-hydrostatic solvers. This restructuring provides deeper physical context for wave evolution dynamics relevant to the ISW simulations in the NSCS.

Furthermore, we have strengthened connections to modern computational tools by explicitly contrasting our MITgcm implementation against recently established community models such as SUNTANS (Zhang et al., 2011) and FVCOM (Lai et al., 2019). These enhancements appear primarily in Section 2.1 (Lines 26-38), where we now demonstrate how our approach aligns with contemporary numerical standards while addressing the specific numerical challenges in the South China Sea. We believe these revisions establish robust theoretical foundations while maintaining appropriate scope for an applied modeling study focused on operational forecasting.

The discussion needs to be cleaned up as some parts read very strangely. There are various tools the authors can use for this. In terms of content, the aforementioned turbulence models are presented as interchangeable when as a point of fact, they are designed for very different things (i.e. Mellor-Yamada versus Gent-McWilliams). There are also strange statements about the source of turbulence that seem at odds with my understanding of ocean physics.

***Response***:

We sincerely appreciate the reviewer's insightful critique regarding the clarity and physical foundations of our turbulence parameterization discussion. We acknowledge that the original text contained ambiguous phrasing that inadvertently conflated distinct turbulence closure approaches while oversimplifying the physical mechanisms driving oceanic mixing. In the revised manuscript, we have undertaken comprehensive revisions in Sections 2.1 and 4.1 to rectify these issues through precise terminology and rigorous physical justification. The revised manuscript now explicitly differentiates between tracer diffusivity and momentum viscosity while clarifying their distinct physical origins. Moreover, detailed discussion on turbulence configuration is now added in Section 5.

Specifically, we emphasize that horizontal eddy viscosity ($A_h$ =1.0×10$^{-2}$ m$^2$ s$^{-1}$) parameterizes unresolved lateral dissipation from inertial ranges and mesoscale processes (Smagorinsky, 1963), while vertical eddy viscosity ($A_v$ =1.0×10$^{-2}$ m$^2$ s$^{-1}$) represents turbulence from shear instabilities and internal wave breaking. This configuration aligns with implementations for marginal China Seas using

MITgcm (Min et al., 2023; Vlasenko et al., 2018). To further validate our viscosity selections, a series of sensitivity experiments in Section 5.1 quantitatively demonstrate that the chosen values optimally preserve ISW characteristics relative to mooring observations. Moreover, vertical diffusivity in ISWFM-NSCS v2.0 is set to $1.0 \times 10^{-5} \, \text{m}^2 \, \text{s}^{-1}$ (Fig. 1), consistent with microstructure measurements of background diapycnal mixing in summer in the northern South China Sea (Shang et al., 2017). Horizontal tracer diffusivity adopts $1.0 \times 10^{-2} \, \text{m}^2 \text{s}^{-1}$, following established subgrid-scale parameterizations for mesoscale-resolving models (Large et al., 1994; Griffies, 2004).

Again, we are grateful for this critique, which has significantly strengthened the methodological rigor of our turbulence treatment.

Finally, I was left wondering how the present methodology compares and contrasts with well-established models like Getm-Gotm when applied to something like the Baltic Sea. The two tools are different in purpose, but it would help the context to contrast them.

***Response*:**

We sincerely appreciate the reviewer's insightful comment regarding the comparison of ISWFM-NSCS with established models like GETM-GOTM (Burchard et al., 2004). While both tools are valuable for oceanographic studies, their design and applicability differ significantly due to distinct scientific objectives and regional dynamics.

GETM-GOTM is a well-validated modeling system tailored for estuarine, coastal, and shelf seas (e.g., the Baltic Sea, North Sea, and Wadden Sea), excelling in resolving baroclinic processes, tidal dynamics, sediment transport, and turbulence closure in shallow, stratified systems (Stips et al., 2004, 2008; Tiessen et al., 2012). Its capabilities in simulating wetting-drying cycles, biogeochemical interactions, and high-resolution coastal processes (e.g., dense bottom currents in the Western Baltic Sea or suspended matter dynamics in the Wadden Sea) make it a powerful tool for coastal regions with complex bathymetry and strong anthropogenic influences.

In contrast, ISWNM-NSCS (built on MITgcm) focuses on deep-water ISW dynamics, particularly in semi-enclosed basins like the South China Sea (SCS). To reproduce large-amplitude features of ISWs propagating over abyssal depths (>2000 m) in the SCS, non-hydrostatic terms are required in the framework. However, GETM-GOTM employs hydrostatic and Boussinesq approximations suitable for shallow systems. We agree that applying ISWNM to the Baltic Sea (a shallow, tidally dominated system) would likely underperform compared to GETM-GOTM, as the latter's explicit treatment of barotropic-baroclinic splitting, wetting-drying, and turbulence closures better aligns with Baltic Sea dynamics. Conversely, ISWNM-NSCS is optimized for deep basins where internal wave energetics dominate over tidal-driven mixing. To clarify this distinction, we have expanded our

discussion in the revised manuscript (Section 5.3) to contrast the two models' strengths, limitations, and regional suitability.

We regard this comparison as a constructive pathway for future work, particularly in exploring hybrid approaches or cross-validation across diverse ocean regimes.

---

## Author Response (AR3)

**Response letter to the referee #3**

The authors have addressed the majority of my concerns and I think the paper is very close to being publication worthy. The one area outstanding is on the theory side. My point was not that one weakly nonlinear theory reference (Gear and Grimshaw) should be replaced by another, but that the whole idea of using weakly nonlinear theory as a quantitative tool is flawed. Between myself and Kevin Lamb there have been at least ten papers showing the various ways in which KdV type theories fail, and that the exact DJL theory can (sometimes) succeed where they fail. As I mentioned, there is even a monograph I wrote on the open source uses of the DJL theory. So please include something to even out the discussion. It isn't just weakly nonlinear theory or regional models, there is a lot more in between.

***Response*:**

We sincerely thank the reviewer for this valuable comment regarding the theoretical framework discussion. We fully acknowledge that the original manuscript failed to reflect contemporary advances in solitary wave theory. In this revision, we have substantially revised the introduction to directly address these concerns by:

(1) Explicitly stating that weakly nonlinear (KdV-type) theories are fundamentally flawed as quantitative tools for realistic ISW dynamics particularly in the NSCS, supported by citations to Lamb (1999) and Stastna and Lamb (2008) demonstrating errors in phase speed and wave width estimates;

(2) Elevating DJL theory as the rigorous alternative that succeeds where KdV fails, highlighting its exact formulation without weak-amplitude/long-wave approximations;

(3) Clearly positioning modern 3D non-hydrostatic models (MITgcm, SUNTANS, FVCOM) as the essential bridge between theoretical paradigms and operational systems.

Detailed modifications are listed below (Section 1, Lines 26-47):

"Internal solitary wave (ISW) research has historically relied on theoretical frameworks to describe nonlinear wave dynamics. Weakly nonlinear theories, epitomized by the Korteweg-de Vries (KdV) equation (Benney, 1966) and its extensions (Grimshaw et al., 2010), employ asymptotic expansions to decouple vertical structure from horizontal evolution. While providing valuable conceptual insights, these approximations exhibit systematic quantitative deficiencies for large-amplitude ISWs particularly in the northern South China Sea (NSCS), where vertical displacements of ISWs exceed 200 m (Huang et al., 2017; Alford et al., 2015). Specifically, KdV-type theories might overestimate phase speeds and underestimate wave widths (Lamb, 1999; Stastna and Lamb, 2008), limiting their utility as predictive tools. Concurrently, the exact Dubreil-Jacotin-Long (DJL) theory

emerged as a mathematically complete alternative, solving the stratified Euler equations without amplitude or wavelength approximations (Stastna and Legare, 2024). The DJL equation computes ISW structure and propagation speed through an eigenvalue problem that intrinsically accounts for isopycnal displacement effects, providing high-fidelity solutions even for complex stratifications. Nevertheless, both KdV and DJL approaches share inherent constraints that they describe steady-state waves or slow shoaling dynamics (Lamb and Xiao, 2014), but cannot resolve transient 3D processes and define entire ISW lifecycles in realistic oceans.

To overcome these limitations, high-resolution numerical solvers have become indispensable for simulating ISW dynamics. By the early 21st century, two-layer analytical models (Holloway et al., 1997) and depth-averaged 2D hydrostatic approaches (Du et al., 2008) proved inadequate for capturing non-hydrostatic effects and strong nonlinearity in regions like the NSCS. This spurred development of high-resolution 3D non-hydrostatic solvers capable of resolving critical processes including generation, propagation, and dissipation of ISWs (Simmons et al., 2011). Contemporary open-source frameworks like SUNTANS (Zhang et al., 2011), MITgcm (Vlasenko et al., 2005; Alford et al., 2015), and FVCOM (Lai et al., 2019) now enable realistic simulations of ISW generation, propagation, and dissipation through advanced numerical schemes validated against modern observational arrays. These advances form the foundation for our ISWFM-NSCS model, which bridges the gap between theoretical paradigms and operational forecasting in the NSCS basin."